# Lyso-IP: Uncovering Pathogenic Mechanisms of Lysosomal Dysfunction

**DOI:** 10.3390/biom12050616

**Published:** 2022-04-21

**Authors:** Chase Chen, Ellen Sidransky, Yu Chen

**Affiliations:** 1Section on Molecular Neurogenetics, Medical Genetics Branch, National Human Genome Research Institute, National Institutes of Health, Bld 35A, Room 1E623 35 Convent Drive, MSC 3708, Rockville, MD 20892, USA; chase.chen@nih.gov; 2Aligning Science Across Parkinson’s (ASAP) Collaborative Network, Chevy Chase, MD 20815, USA

**Keywords:** lysosomes, lysosome isolation, Niemann-Pick type C

## Abstract

Lysosomes are ubiquitous membrane-bound organelles found in all eukaryotic cells. Outside of their well-known degradative function, lysosomes are integral in maintaining cellular homeostasis. Growing evidence has shown that lysosomal dysfunction plays an important role not only in the rare group of lysosomal storage diseases but also in a host of others, including common neurodegenerative disorders, such as Alzheimer disease and Parkinson disease. New technological advances have significantly increased our ability to rapidly isolate lysosomes from cells in recent years. The development of the Lyso-IP approach and similar methods now allow for lysosomal purification within ten minutes. Multiple studies using the Lyso-IP approach have revealed novel insights into the pathogenic mechanisms of lysosomal disorders, including Niemann-Pick type C disease, showing the immense potential for this technique. Future applications of rapid lysosomal isolation techniques are likely to greatly enhance our understanding of lysosomal dysfunction in rare and common neurodegeneration causes.

## 1. Introduction

Lysosomes are ubiquitous membrane-bound cytoplasmic organelles that are present in eukaryotic cells and are involved in the degradation of a variety of biological macromolecules [1,2,3]. The lysosomal lumen contains more than 60 acid hydrolases that can break down proteins, carbohydrates, lipids, and nucleic acids [4]. Once the macromolecules are degraded, their building blocks are then recycled within the cell. In addition to their catabolic function, lysosomes play important roles in the regulation of physiological processes within the cell, such as cellular signaling, cholesterol homeostasis, bone and tissue remodeling, and plasma membrane repair [5]. The mammalian lysosome also contains more than 25 integral transmembrane proteins that regulate the acidification of the lysosomal lumen, protein transport, membrane fusion, autophagy, and lysosomal biogenesis [6].

Dysfunction of lysosomal hydrolases is associated with a class of rare diseases known as lysosomal storage disorders (LSDs) [7]. As a group, LSDs include over 50 monogenic disorders of lysosomal catabolism, the majority of which are inherited in an autosomal recessive pattern. Although individual LSDs are rare, the collective incidence of LSDs is estimated to be around one in five thousand individuals [7,8]. The clinical manifestations of LSDs are extremely broad, but many LSDs present in early childhood and can have progressive neurodegenerative and diverse systemic manifestations [9]. While our overall understanding of the LSDs, including treatments and therapies, has dramatically improved in recent years, the cellular pathogenesis of many LSDs is still poorly understood.

The purification of lysosomes is an important tool to facilitate elucidating lysosomal dysfunction in LSDs. To further understand the function and regulation of lysosomal pathways, it is critical to study isolated lysosomes in order to detect subtle changes in the lysosomal composition and the presence of low-abundance molecules. The advance of analytical techniques and the establishment of large-scale omics studies over the past few years have made the need for analyses of isolated organelles even more critical [10]. Analysis of lysosomal proteins, metabolites, and lipids can now yield important insights into relevant disease mechanisms. Since Christian De Duve first identified the lysosome as an organelle in 1955 through centrifugal fractionation, a multitude of different techniques for lysosomal purification have been developed [11]. In this review, we will discuss recent advancements in the purification of lysosomes and how these tools can be used to improve our collective understanding of lysosomal function in the context of LSDs and beyond.

In addition to the LSDs, there is an increasing awareness of the role of the lysosome in the pathogenesis of a gamut of diverse common and rare disorders. Specifically, genes regulating lysosomal function are now implicated in common neurodegenerative disorders, such as Parkinson disease (PD), Alzheimer disease (AD), and amyotrophic lateral sclerosis (ALS) [12,13].

## 2. Traditional Methods for Lysosome Isolation

Historically, the most common strategy for lysosome isolation has involved some variation of density gradient differential centrifugation (Table 1), where different organelles (e.g., exosomes, mitochondria, and lysosomes) from whole cell lysates are separated by differences in density [14]. Cells are first mechanically lysed, followed by successive rounds of centrifugation with increasing force and duration aimed to separate particles by size, with larger particles being sedimented first. The addition of a density-based gradient (e.g., sucrose, iodixanol, Percoll, or cesium chloride) helps to further separate subcellular organelles according to their individual densities. A major drawback of density gradient centrifugation methods is the inability to isolate relatively pure fractions of lysosomes due to the heterogeneity of densities across organelles. Often, isolated lysosomal fractions will be contaminated with mitochondria and peroxisomes [15,16]. To avoid this problem, studies have shown that the addition of the detergent Triton WR1339 can lead to a selective decrease in the equilibrium density of lysosomes by forming “tritosomes”, thus improving separation [16,17]. However, lysosomes purified using this method contain a large quantity of non-physiological material that can alter the lipidome and metabolome. Furthermore, differential centrifugation methods require expensive ultra-high-speed centrifuges, large quantities of starting material, and multiple lengthy centrifugation steps (e.g., 141,000× *g* for 90 min) [18]. The extended duration of these centrifugation protocols often leads to lysosomal rupture and the loss of labile molecules.

Alternative strategies, such as Fluorescence-Assisted Organelle Sorting (FAOS), have been developed to purify lysosomes. FAOS relies on the same principles of Fluorescence-Activated Cell Sorting (FACS), where fluorescently labeled lysosomes are isolated as they pass through a detection laser separating fluorescent lysosomes from other organelles [28]. Using FAOS, one study successfully isolated lysosomes from a stably transfected mast cell line expressing a fluorescently tagged rat mast cell protease (RMCP-DsRed) that localized to secretory lysosomes efficiently [29]. The sorted lysosomes showed a 50% increase in specific activity of the lysosomal enzyme hexosaminidase [29]. However, the major disadvantages of the FAOS technique are the potential for individual lysosomes to break open during the sorting process, and it can be incredibly difficult to sort organelles that are orders of magnitude smaller than whole cells [10].

Other widely used lysosomal isolation methods rely on magnets to purify iron-laden lysosomes. In this strategy, iron-dextran particles are taken up by the cells and are delivered to lysosomes through endocytic pathways. Then, iron-containing lysosomes are isolated with a magnet [30]. More recently, further technological advancements have led to the development of smaller superparamagnetic iron oxide nanoparticles (SPIONs) [31]. SPIONs are nanoparticles consisting of an inorganic magnetic core surrounded by an organic or inorganic shell that helps provide stability and improve functionality. Such nanoparticles have already been used for a variety of biomedical applications. Due to their small size, SPIONs can enter cells through regular endocytic pathways and easily localize to the lysosome. In addition, studies show that SPIONs coated with dimercaptosuccinic acid (DMSA) can specifically target late endosomes and lysosomes [31]. One study successfully used SPIONs to purify lysosomes from Niemann-Pick disease type C1 (NPC1)-deficient HeLa cells for omics-based profiling [31]. Although the development of SPIONs is a significant improvement over iron-dextran-based lysosomal purification, the introduction of iron to the cell can potentially disrupt the native cellular environment of the lysosome. Additionally, both the process of producing DMSA-coated SPIONs and the lysosome isolation protocols are labor-intensive, lengthy, and require the use of dangerous and highly corrosive acidic solutions [20].

Affinity purification and immunoisolation strategies for lysosomal isolation have also been developed. One immunoisolation method specifically targeted the vacuolar H^+^-ATPase (V-ATPase) pump [21]. The V-ATPase pump resides in the limiting membrane of the lysosome and is responsible for acidifying maturing endosomes. Using multiple antibodies targeting the A or B domain of the V_1_ subunit of the V-ATPase pump, pure lysosomes suitable for omics analysis were effectively isolated [21]. However, this strategy still requires a lengthy 90-min incubation with V-ATPase antibodies, uses a sucrose extraction buffer that interferes with downstream mass spectrometry (MS) analysis, and uses large quantities of multiple antibodies.

## 3. The Development of Lyso-IP and Similar Lysosomal Isolation Techniques

Over the past few years, a novel method for lysosome purification, Lyso-IP, has been developed. With the Lyso-IP technique, intact lysosomes are immunoprecipitated via epitopes that are added to the cytosolic portion of lysosomal transmembrane proteins [27]. First published by Abu-Remaileh et al., the fusion of triple human influenza virus hemagglutinin (3×HA) epitope tags to the lysosomal transmembrane protein 192 (TMEM192), TMEM192-3×HA, allowed for the rapid isolation of lysosomes by using anti-HA magnetic beads. Remarkably, this takes only approximately ten minutes to accomplish. The isolated lysosomes retained lysosomal protease activity and contained many lysosomal luminal proteins [27]. TMEM192-3×HA is a significant improvement over earlier attempts at this strategy, where the lysosome-associated membrane protein 1 (LAMP1) fused to a cytosolic FLAG epitope tag (LAMP1-FLAG) failed to purify lysosomes efficiently [32]. Additionally, the rapid timeframe of the TMEM192-tagged Lyso-IP method allows potentially labile molecules, such as amino acids and other metabolites, to remain stable for downstream MS analysis. Lysosomes isolated by the Lyso-IP method have been successfully used for proteomic, lipidomic, and metabolomic analyses [22,26,27].

The Lyso-IP approach was adapted from a similar method for rapid mitochondrial isolation (Mito-IP), which utilized the mitochondrial transmembrane protein OMP25 fused to triple HA epitope tags [33]. The advantages of Mito-IP include the use of a “KPBS” buffer (136 mM KCl, 10 mM KH_2_PO_4_, pH 7.25) that is LC-MS compatible and stabilizes the integrity of intact mitochondria [33,34]. The KPBS buffer functions to mimic cytosolic potassium concentrations supporting the mitochondrial membrane potential. Other small molecules (e.g., sucrose) can interfere with downstream LC/MS analysis, while sodium-based buffers can disrupt the mitochondrial membrane potential [35]. Additionally, the size of the antibody-conjugated magnetic beads is crucial for successful organelle immunoprecipitation. Small (~1 μM) cognate non-porous magnetic beads, such as anti-HA magnetic beads, provide better yields of proteins and metabolites [33], whereas large magnetic beads with a porous matrix (e.g., FLAG) trap organelles, leading to poor yields.

Alterations to the Lyso-IP and Mito-IP methods have been made, specifically taking advantage of the high affinity twin strep tag and streptavidin variants [36]. Compared to the micromolar-to-nanomolar dissociation constants (K_d_) of commercial epitope tags (e.g., HA and FLAG), the twin strep tag has a K_d_ in the nanomolar-to-picomolar range [37]. The smaller K_d_ of the twin strep tag allows for more rapid purification and requires less starting material to yield identical amounts of purified organelles. In addition, the twin strep tag retains the ability to be eluted by low concentrations of biotin while preserving the native biotin and streptavidin interaction. The twin strep tag can also be broadly applied to purify mitochondria, lysosomes, and peroxisomes and permits purification of different organelles from the same sample [36].

Systematic comparison of multiple strategies for lysosome purification (Figure 1), including organelle-enriched pellet, two-step sucrose density-based centrifugation, SPIONs, and Lyso-IP, showed that SPIONs and Lyso-IP methods yielded significantly better coverage of the lysosomal proteome [38]. Compared to organelle-enriched pellet isolation (centrifugation of post-nuclear supernatant at 20,000× *g*), SPIONs and Lyso-IP methods showed as high as a 118-fold increase in the enrichment of specific lysosomal protein markers. In addition, compared to whole cell lysates, the lysosomal fractions from SPIONs and Lyso-IP showed a 12% increase in known lysosomal proteins, whereas no increase was found with the other methods [38]. These results suggest that isolation of lysosomes by SPIONs and Lyso-IP present the best methods for label-free quantification data.

Furthermore, when comparing the efficiency of SPIONs and Lyso-IP, it was found that the Lyso-IP method results in a higher number of intact lysosomes but with an overall lower yield. The potential loss of lysosome-associated and luminal proteins during the Lyso-IP purification process led to a greater variation in protein abundance. Thus, it was concluded that isolation with SPIONs gave the highest number of lysosomes per starting material and preserved the integrity of the lysosomes most efficiently [38]. However, this analysis was limited in scope to the lysosomes of HEK293 cells only. More work is needed to further optimize the protocols and validate the efficacy of lysosome purification strategies, such as SPIONs and Lyso-IP, in different cellular models.

## 4. Novel Insights into the Pathogenic Mechanisms of Niemann-Pick Type C Disease Revealed by the Lyso-IP Approach

The application of the Lyso-IP approach to the field of LSD has uncovered critical insights into the mechanisms of lysosomal dysfunction in the LSD Niemann-Pick type C (NPC). NPC results from mutations in the *NPC1* and *NPC2* genes, which leads to the accumulation of endocytosed intra-lysosomal cholesterol [39,40]. It was first noted that cholesterol, an essential building block for cellular growth, drives mTORC1 recruitment and activation at the lysosomal surface [22]. Additional studies completed in NPC-cellular models have shown mTORC1 is aberrantly hyperactive when *NPC1* is knocked out [23]. To determine the mechanism underlying constitutive mTORC1 activation in NPC models, the protein composition of Lyso-IP lysosomes was analyzed by proteomic evaluations. This analysis identified oxysterol binding protein (OSBP) as the cholesterol carrier that delivers cholesterol across ER–lysosome contact points to the limiting membrane of lysosomes, leading to mTORC1 activation [41]. In cells lacking NPC1, cholesterol accumulates both in the lysosomal lumen and by unopposed ER-to-lysosome transport via OSBP on the lysosomal membrane. In turn, excess lysosome-membrane cholesterol constitutively activates mTORC1 signaling and inhibits autophagy.

When proteomic profiling of control and NPC-lysosomes was performed using the Lyso-IP approach, pronounced proteolytic impairment was documented for NPC lysosomes, accompanied by hydrolase depletion, increased membrane micro-damage, accumulated autophagic adaptors, and defective mitophagy [23]. Importantly, it appears that hyperactive mTORC1 activity, rather than cholesterol buildup, is what drives these dysfunctions because pharmacologic and genetic inhibition of mTORC1 was able to correct compositional and functional defects of NPC lysosomes while not altering cholesterol accumulation. The restorative effects of mTORC1 inhibition on lysosomal function suggest that manipulating this pathway could have therapeutic value in NPC and other LSDs where hyperactive mTORC1 is present.

## 5. The Lyso-IP Approach Yields Insights into Other Lysosomal Functions

Lyso-IP has also uncovered insights into the amino acid and lipid exchange mechanisms between lysosomes and cytosol. We now know that efflux of amino acids and lysosomal metabolites are controlled by the V-ATPase pump and mTOR dependent mechanisms [27]. Furthermore, the small GTPase Rab32 was identified as an important regulator of cellular metabolism, controlling the master regulator of cell growth, mechanistic Target of Rapamycin Complex 1 (mTORC1) activity [42]. Additionally, Lyso-IP studies have shown that activation of mTORC1 is linked to lysosomal arginine sensing by the lysosomal transmembrane protein SLC38A9 interacting with the Ragulator–Rag complex [24]. These works present just a fraction of the potential insights into lysosome regulation we can gain from the Lyso-IP method.

Proteomic profiling of Lyso-IP lysosomes has also increased our understanding of autophagy-related mechanisms. Quantitative proteomics profiling of Lyso-IP lysosomes identified the protein NUFIP1, which was not previously associated with lysosomes [26]. Under starvation conditions, NUFIP1 localizes to autophagosomes and lysosomes and acts as an autophagy receptor for ribosomes. NUFIP1 binds directly to LC3B to induce the autophagy of ribosomes (ribophagy) and is responsible for promoting cell survival [26].

In addition to cellular models, the Lyso-IP method can easily be applied to different animal models of disease. The generation of MITO-tag mice has already improved our understanding of mitochondrial physiology by providing a simple platform for assessing the mitochondrial metabolism in vivo [43]. Likewise, introducing the LYSO-tag into different mouse models of disease could contribute to similar successes for our understanding of lysosomal metabolism, and the development of such a model is currently ongoing.

## 6. Conclusions

Accumulating data from cellular models utilizing the Lyso-IP method has already contributed significantly to our understanding of lysosomal regulation, particularly regarding mTORC1 and NPC regulation. Furthermore, quantitative proteomic profiling of Lyso-IP lysosomes has expanded our knowledge regarding the nutrient exchange mechanisms between the lysosome and the cytosol. In addition, new regulators and lysosome-associated proteins (e.g., Rab32) have been discovered in part due to the Lyso-IP method. These results only appear to be the tip of the iceberg and further applications of this technique in studies of different LSDs are already underway. While some engineering is required upfront to create cells expressing the Lyso-tag, ultimately, the rapid protocol and ease of isolation makes the Lyso-IP a particularly attractive method for isolating pure lysosomes. Further optimizations and applications of the Lyso-IP method, especially to animal models, could vastly improve our understanding of lysosomal dysfunction in the LSDs and beyond. Moreover, the technique may yield exciting advances in the field of neurodegenerative disease, catapulting investigations into how lysosomal dysfunction contributes to the pathogenesis of these devastating and common disorders.

## Figures and Tables

**Figure 1 biomolecules-12-00616-f001:**
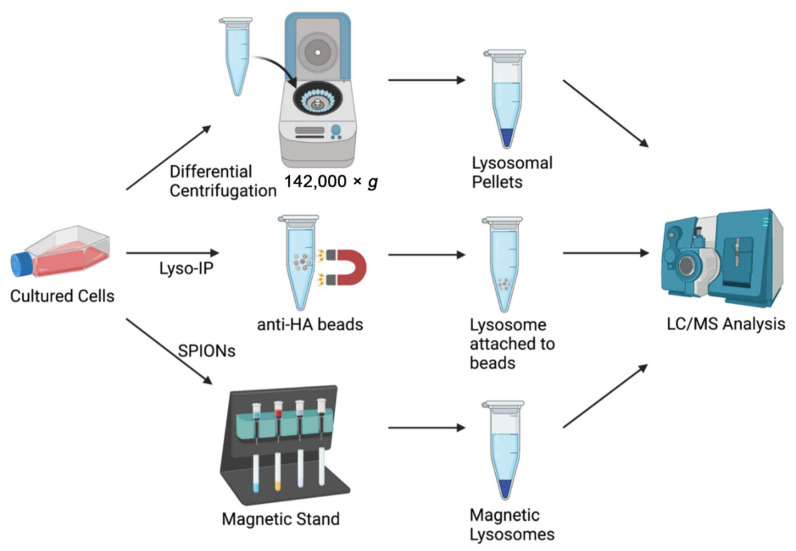
Widely used techniques for lysosome isolation include differential centrifugation (e.g., 142,000× *g* for 90 min), Lyso-IP (immunopurification of lysosomal transmembrane protein with cytosolic HA-tag), and SPIONs (magnetic isolation of DMSA-coated SPIONs).

**Table 1 biomolecules-12-00616-t001:** Summary of the most common laboratory techniques for lysosomal isolation from eukaryotic cells. Detailed protocols can be found in the Methods section of each reference.

Isolation Method	Advantages	Disadvantages
Density Gradient Differential Centrifugation [18]	Commonly used, well-established methods	High chance of contamination with other organelles (e.g., mitochondria and peroxisomes), long and laborious protocols, requires ultra-high-speed centrifuge
Fluorescence-Assisted Organelle Sorting (FAOS) [19]	High specificity and efficiency	Easy to break open organelles during sorting, difficult to sort small organelles (e.g., lysosomes), requires overexpression of fluorescently labeled lysosomal protein
Superparamagnetic Iron Oxide Nanoparticles (SPIONs) [20]	Higher yield, preserves integrity of lysosomes	Labor-intensive protocol to generate DMSA-coated SPIONs, requires use of dangerous chemicals, modifies native lysosomal environment
Affinity Purification and Immunoisolation [21]	No modifications of lysosome, efficient isolation	Long incubation times, requires large amounts of specific antibodies
Lyso-IP [22,23,24,25,26,27]	Rapid isolation of intact lysosomes (~10 min), uses LC/MS compatible buffer, preserves labile molecules, applicable to cellular and animal models, preserves integrity of lysosomes	Lower yield compared to SPIONs, higher variability in lysosomal proteins detected, requires overexpression of lysosomal transmembrane protein, not possible in human tissue samples

## Data Availability

Not applicable.

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
