# Peer review of "Lyso-IP: Uncovering Pathogenic Mechanisms of Lysosomal Dysfunction"

_biomolecules, 2022, doi:10.3390/biom12050616_

Round 1
Reviewer 1 Report
Biomolecules-1677289
Chen et al.
In this manuscript the authors review the technique of Lyso-IP, an emerging tool for interrogating lysosomal mechanisms and disease. They discuss the development of the methods and compare to earlier technologies. As an example, they highlight the utility of the Lyso-IP approach in the context of Niemann-Pick C disease. They conclude by discussing future applications.
The review is clearly written and logically presented, and provides an excellent review of alternate methods for lysosomal isolation. If there is a limitation, it is that the technique has primarily been used by a limited number of labs and applied in only a few situations. In that sense, the review is timely and will help in dissemination of an important new methodology.
Specific points:
- A drawback of the tritosomes is not just that there is an abundance of the non-physiologic lipid in the lysosomes, but that the presence of this lipid will alter the lipidome and metabolome.
- The authors correctly point out the cons of the SPIONs approach, including the limitation to HEK293 cells. However, this is also an issue with Lyso-IP. The method has been optimized for HEK293 cells and will require work on the part of others to adapt the protocol to different cell lines.
Minor points:
Line 15: use singular for Alzheimer
Line 180: typo “leading”
Line 227: A better reference for NPC1 and NPC2 cellular phenotypes is Vanier MT, J Inherited Metabolic Disease 2015 38:187-99
Author Response
Response to Reviewer 1
In this manuscript the authors review the technique of Lyso-IP, an emerging tool for interrogating lysosomal mechanisms and disease. They discuss the development of the methods and compare to earlier technologies. As an example, they highlight the utility of the Lyso-IP approach in the context of Niemann-Pick C disease. They conclude by discussing future applications.The review is clearly written and logically presented, and provides an excellent review of alternate methods for lysosomal isolation. If there is a limitation, it is that the technique has primarily been used by a limited number of labs and applied in only a few situations. In that sense, the review is timely and will help in dissemination of an important new methodology.
Specific points:
- A drawback of the tritosomes is not just that there is an abundance of the non-physiologic lipid in the lysosomes, but that the presence of this lipid will alter the lipidome and metabolome.
Response: Thank you for this comment. On line 92 we now state
“However, lysosomes purified using this method contain a large quantity of non-physiological material that can alter the lipidome and metabolome.”
- The authors correctly point out the cons of the SPIONs approach, including the limitation to HEK293 cells. However, this is also an issue with Lyso-IP. The method has been optimized for HEK293 cells and will require work on the part of others to adapt the protocol to different cell -kip lines.
Response: In line 217 we now state
“More work is needed to further optimize protocols and validate the efficacy of lysosome purification strategies, such as SPIONs and Lyso-IP, in different cellular models”.
Minor points:
Line 15: use singular for Alzheimer
Line 180: typo “leading”
Response: These errors were corrected
Line 227: A better reference for NPC1 and NPC2 cellular phenotypes is Vanier MT, J Inherited Metabolic Disease 2015 38:187-99
Response: This is now reference #40.
Reviewer 2 Report
In their review article, Chen et al. summarize and compare five methods that are currently used for the isolation of lysosomes, i.e. density gradient differential centrifugation, FAOS, SPIONs, affinity purification, and Lyso-IP. Their special focus is on the newly developed method Lyso-IP. For this method, the authors also give examples where this method has been used in cell models of the lysosomal storage disorder Niemann Pick Type C and in the study of lysosomal functions.
The review gives a nice overview on the pros and cons of the different methods, but does not describe the methods in detail.
Minor comments:
- Line 80: the authors mention sucrose and cesium chloride as components of density-gradients. They could also mention more common substances such as Iodixanol (Optiprep) or Percoll
- Table 1: the FAOS method also requires overexpression of a labeled lysosomal protein which is a disadvantage of this method
- A detailed step-by-step protocol for each of the 5 methods or at least for the Lyso-IP could be included (in the supplementary)
Author Response
In their review article, Chen et al. summarize and compare five methods that are currently used for the isolation of lysosomes, i.e. density gradient differential centrifugation, FAOS, SPIONs, affinity purification, and Lyso-IP. Their special focus is on the newly developed method Lyso-IP. For this method, the authors also give examples where this method has been used in cell models of the lysosomal storage disorder Niemann Pick Type C and in the study of lysosomal functions.The review gives a nice overview on the pros and cons of the different methods, but does not describe the methods in detail.
Minor comments:
- Line 80: the authors mention sucrose and cesium chloride as components of density-gradients. They could also mention more common substances such as Iodixanol (Optiprep) or Percoll
Response: We now state: “The addition of a density-based gradient (e.g. sucrose, iodixanol, Percoll, or cesium chloride) helps to further separate subcellular organelles according to their individual densities.”
- Table 1: the FAOS method also requires overexpression of a labeled lysosomal protein which is a disadvantage of this method
Response: In the table, under disadvantages, we now specify that the FAOS method requires overexpression of labeled lysosomal protein
- A detailed step-by-step protocol for each of the 5 methods or at least for the Lyso-IP could be included (in the supplementary)
Response: Since this is a review of the work of others, we have chosen not to include protocols that we have not validated in our own hands, However in Table 1 we specifically state that detailed protocols can be found in the Methods section of each cited reference.